# Age at natural menopause and its determinants in female population of Kharameh cohort study: Comparison of regression, conditional tree and forests

Zahra Pasokh[1], Mozhgan Seif[2]*, Haleh Ghaem[2], Abbas Rezaianzadeh[3], Masoumeh Ghoddusi Johari[4]

1 Student Research Committee, Shiraz University of Medical Sciences, Shiraz, Iran, 2 Non-Communicable Diseases Research Center, Department of Epidemiology, School of Health, Shiraz University of Medical Sciences, Shiraz, Iran, 3 Colorectal Research Center, Shiraz University of Medical Sciences, Shiraz, Iran, 4 Breast Diseases Research Center, Shiraz University of Medical Sciences, Shiraz, Iran

* mozhgan.safe@gmail.com

## Abstract

### Background

Natural menopause is defined as the permanent cessation of menstruation that occurs after 12 consecutive months of amenorrhea without any obvious pathological or physiological cause. The age of this phenomenon has been reported to be associated with several health outcomes.

### Objectives

This study aimed to estimate the Age at Natural Menopause (ANM) and to identify reproductive and demographic factors affecting ANM.

### Methods

This cross-sectional, population-based study was conducted on 2517 post-menopausal women aged 40–70 years participating in the first phase of the PERSIAN cohort study of Kharameh, Iran, during 2014–2017. To more accurately detect the determinants of ANM, we applied multiple linear regression beside some machine learning algorithms including conditional tree, conditional forest, and random forest. Then, the fitness of these methods was compared using Mean Squared Error (MSE) and Pearson correlation coefficient.

### Results

The mean±SD of ANM was 48.95±6.13. Both applied forests provided more accurate results and identified more predictors. However, according to the final comparison, the conditional forest was the most accurate method which recognized that more pregnancies, longer breastfeeding, Fars ethnicity, and urbanization have the greatest impact on later ANM.

**Data Availability Statement:** Data cannot be shared publicly because data contain potentially

identifying and sensitive patient information. Data are available from the kharameh cohort center (contact via: kheramehcohort@gmail.com) for researchers who meet the criteria for access to confidential data.

**Funding:** This study was supported by the Research Vice-chancellor of Shiraz University of Medical Sciences (grant No. 25007). The funders had no role in study design, data collection and analysis, decision to publish, or preparation of the manuscript.

**Competing interests:** The authors have declared that no competing interests exist.

## Conclusions

This study found a wide range of reproductive and demographic factors affecting ANM. Considering our findings in decision-making can reduce the complications related to this phenomenon and, consequently, improve the quality of life of post-menopausal women.

## Introduction

Age at the onset of menopause is the most important determinant of women's future health outcomes [1]. During the aging process, the number and quality of the oocytes in the ovaries decrease, the associated cyclic endocrinological activities stop, and consequently, women enter menopause [2]. Natural menopause corresponds to the last menstrual period (LMP) and is recognized to have occurred after 12 consecutive months of amenorrhea for which there is no other obvious pathological or physiological cause [3].

The median age of the world's population is increasing due to the decline in fertility and the increase in life expectancy [4]. Accordingly, most women spend more than a third of their lives in the postmenopausal period. The timing of menopause is claimed to be a substantial risk factor for some chronic diseases. Previous studies have demonstrated that lower ANM is associated with cardiovascular disease [5], all-cause mortality [6], worse cognitive function [7], type 2 diabetes [8], osteoporosis, and fracture [9]. In contrast, higher ANM is associated with breast cancer [10], and uterine/ovarian cancer [11]. These complications highlight the importance of identifying factors that can affect ANM.

Globally, the average age of menopause is estimated at 50 years [12]. Several studies have indicated that women living in developing countries, experience natural menopause several years earlier than those in developed countries [13]. According to a systematic review and meta-analysis performed in 2017, the average age of menopause in Iran was estimated as 48.57 years [14] which is lower than Europe (50.54), Australia (51.25), and the USA (49.11) but higher than the Middle East (47.37), Latin America (47.24) and almost equal to Asia (48.75) and Africa (48.38) [15].

ANM substantially varies across different ethnicities, regions, and countries due to genetic background, socioeconomic position, environment, lifestyle, and reproductive or early childhood factors [15]. Many studies have been conducted around the world to investigate the determinants of menopause onset, and Iran is no exception. They have indicated that demographic factors such as socioeconomic status, occupation, education, residence, smoking, and some reproductive factors like breastfeeding, menstruation start age, marital status, pregnancy, and other factors are related to ANM [16–24] but whether these factors are definitive in all women or even increase or decrease the age of menopause remain controversial. For instance, in Farjam et al.'s study, menopausal age was lower in not-married, employed and illiterate women [19] while in another study conducted by Vatankhah et al., Education, employment, and marital status were not associated with early menopause [25]. Although in several studies, women with lower socioeconomic status had a younger menopausal age [20, 23], in some other studies, it wasn't associated with menopausal age [18]. Therefore, it is difficult to predict the ANM due to these conflicting results. In other words, the impact of many of these predictors on ANM may be influenced by racial, ethnic, and environmental characteristics and specific to a particular geographical and racial area. Furthermore, to date, there are a few studies that have used machine learning algorithms to determine the factors influencing the age of menopause.

This study aimed to estimate the ANM and to identify reproductive and demographic factors affecting ANM. Therefore, we used some machine learning algorithms including

conditional tree, random and conditional forest beside multiple linear regression as a basic method hoping that by more precisely identifying the determinants of the ANM, a step will be taken to prevent its complication and thereby promote the health and the quality of life of menopausal and postmenopausal women.

## Materials and methods

This cross-sectional, population-based study was conducted on 2517 menopausal women aged 40–70 years who were enrolled from 2014 to 2017 in the first phase of the PERSIAN (Prospective Epidemiological Research Studies in Iran) cohort study of Kharameh County (located in Fars Province in the southern part of Iran).

The PERSIAN cohort study was launched in 2014 by the Ministry of Health and Medical Education to identify the most prevalent Noncommunicable Diseases (NCDs) among Iran's ethnic groups and to investigate effective methods of prevention [26]. The PERSIAN Cohort Study started in Kharameh in October 2014, and enrollment ended in February 2017.

It should be declared that in the Kharameh cohort study, eligible participants were entered through the census. After obtaining the consent of the participants, standardized questionnaires containing reproductive and demographic information were completed by trained staff through face-to-face interviews. The validity and reliability of the questionnaires applied in this study had been already checked [26].

In the Kharameh cohort study, the whole residents of Kharameh, aged 40–70 years with Iranian nationality were invited to participate in this study. Some people were excluded from the study due to not attending the clinics for physical examination or mental retardation or unwillingness to participate in the study. Eventually, out of 10,836 people, 10,663 people (5,944 women and 4,719 men), which is equivalent to 98.4% of the population of Kharamah, participated in this census. Data collected in the Kharameh cohort study were accessed for our study in February 2022 after approval by the research committee. Since this study aims to determine associated factors with the ANM, we used the information of women who reported having natural menopause. In other words, we refused to use the information of women who had induced menopause due to hysterectomy, bilateral oophorectomy, illness, or other causes. Out of 5944 women, 2517 people were reported to be naturally menopause.

This study received approval from the Ethics Committee of Shiraz University of Medical Sciences (IR.SUMS.SCHEANUT.REC.1400.104). Informed consent was obtained from all participants prior to their involvement. The participants were provided with assurance regarding confidentiality and anonymity.

In the current survey, we utilized several main reproductive and demographic information collected from the Kharameh cohort study to predict the ANM. Reproductive factors included marital status (single, married, divorced, widowed, others), menstruation start age, age at first marriage, number of pregnancies, history of abortion and stillbirth, breastfeeding duration, tubectomy, use of contraceptive drugs, and hormonal replacement drugs. Demographic factors included socioeconomic status (SES) which is calculated using the principle component analysis method by considering the following variables: homeownership, home size, number of bathrooms in the house, owing cars and their prices, domestic and international travels, reading books, access to the internet, owning mobile phones, computers, televisions, washing machines, dishwashers, refrigerators, vacuum cleaners, and microwaves. The resulting index is categorized into four levels (low, moderate, high, and very high). Other demographic variables are as follows: education years, residence (city or village), job (employed or housewife), and mother and father's ethnicity (Fars or other ethnicities).

In this study, the data were analyzed using the R statistical software, version 4.2.0. To recognize the probable factors associated with ANM and also to determine the importance of these factors, we employed multiple linear regression (using the "stepwise" method) and machine learning methods through a recursive partitioning algorithm consisting of the conditional tree (using the "party" package), conditional forest (using the "party" package) and random forest (using the "randomForestSRC" package). In this study, the level of significance was 0.05.

Random and conditional forests are ensemble methods that combine a large collection of trees. The main difference between random forests (Rforest) and conditional forests (Cforest) is the different base learners exerted by the two methods. Rforests are built from classification and regression trees (CART trees), while Cforests are made from conditional inference trees (Ctree) [27]. It can be stated that the greater number of trees in the forest leads to lower variance and higher accuracy.

When fitting a Ctree, the conditional distribution of the statistics that measure the associations between the response variable and the predictor variables is driven. Multiple testing procedures (adjusted P-values) can be used to define whether there are statistically significant associations between any of the predictors and the response. If present, the predictor variable that has the strongest association with the response variable is selected for splitting; otherwise, the recursive procedure will stop. However, when fitting the CART trees, at each node, a small random subset of predictor variables could be chosen and the best split over these variables which causes the purest node with the lowest MSE will be found. Indeed, the splitting procedure is repeated until a certain stopping criterion (the node is pure or its size is smaller than a pre-specified value) is met. Generally, the tree is grown until it is adequately large, without pruning [27]. However, pruning is less important for trees that form a forest; because a forest is an ensemble of trees and a limited number of over-fitted trees do not change the main result of the forest which is an aggregation over all trees. Therefore, the trees of a forest are usually unpruned.

Another property of the forest is the number of candidate variables for splitting in each node. As mentioned, in the growth of a tree, it is possible to first select a random subset of variables and then after testing the selected variables, the splitting is done on the most suitable variable. However, this property, like pruning, is less important in trees of a forest than the fitness of one tree and remains controversial. It is shown that the performance of forests is insensitive to the number of predictors considered at each split [28], and its results are typically near optimal over a wide range of the number of candidate variables for splitting [29]. Breiman suggested that a third of the total number of predictors should be randomly selected for regression trees of a forest, however, the square root of the number of predictors should be used if a forest is consisted of classification trees [30].

The result of a forest is usually reported as the importance of variables. In this study, Breiman-Cutler permutation VIMP is used, which is the decrease of prediction accuracy in conditions where the observations are permuted over the interested variable. Simply, the importance of a variable is the difference in the accuracy of the prediction obtained from the following two forests:

1. A forest that grows normally with the real data set.

2. A forest that grows with real data but is permuted on the interested variable.

Although other methods have been proposed to measure the importance of a variable in the forests; For example, decrease in the prediction accuracy due to the random allocation to the daughter node in a situation where splitting has occurred on the interested variable.

In this study, the forests consisted of 1,000 trees, and the number of candidate variables for splitting was considered one-third of the total number of predictors. For tree growing, due to the small number of predictors, especially in comparison to the number of observations, all variables were candidates as split variable at each node split.

To evaluate the performance of the models, we split the dataset into a training set (containing 70% of the data) and a test set (containing 30% of the data). The training set was used to train the models. The models fit and learn from this data. The performance of these models was measured using the test set. Finally, the goodness of fit of the models was assessed by calculating and then comparing the MSE and Pearson correlation coefficient between predicted and observed values.

## Results

This study was conducted on 2517 menopausal women aged 40 to 70 years. The mean±SD (95%CI) and median (1st Qu, 3rd Qu) of ANM were 48.95±6.13 (48.71, 49.19) and 50.00 (45.00,53.00) years, respectively. Table 1 shows the reproductive and demographic factors associated with ANM via regression model. As can be seen, increasing the socioeconomic level would increase the ANM by 0.45 year (p-value <0.001). Urban women who contain 31% of the population, had an average 0.67 year higher ANM than rural ones (p-value = 0.027). In Kharameh, 78% of women were housewives who had 0.70 years higher ANM than employed ones (p-value = 0.019). The people of Kharameh are mostly Fars. So, we divided the ethnicity of women's mother and father into 2 groups: Fars and other ethnicities. In this study, other ethnicities are Arab, Turk, Lor, Azeri, and Baluch. But each group consists of a small number of people. Therefore, we considered all of them as other ethnicities. Regression analysis revealed that only the mother's ethnicity is related to ANM. According to the mother's ethnicity, Fars women, with 68% of the population, had an average 0.98 year higher ANM than

**Table 1. Identified factors associated with menopause age using multiple linear regression among postmenopausal women of Kharameh.**

| Qualitative variables | Categories | Count | Mean ± SD of menopause age | B | S.E | P value |
|---|---|---|---|---|---|---|
| Hormonal replacement drugs | Yes | 47 | 46.723±8.541 | -1.945 | 0.884 | 0.028 |
| | No | 2470 | 48.991±6.073 | | | |
| Pregnancy number | < = 7 | 1314 | 48.239±5.853 | 1.065 | 0.292 | <0.001 |
| | >7 | 1203 | 49.724±6.337 | | | |
| Stillbirth | Yes | 471 | 48.442±6.528 | -0.876 | 0.315 | 0.005 |
| | No | 2046 | 49.065±6.034 | | | |
| Socioeconomic status | low | 804 | 48.361±6.459 | 0.454 | 0.129 | <0.001 |
| | moderate | 826 | 48.981±6.270 | | | |
| | high | 603 | 49.420±5.558 | | | |
| | very high | 284 | 49.521±5.827 | | | |
| Residence | City | 772 | 49.388 ± 5.982 | 0.667 | 0.301 | 0.027 |
| | Village | 1745 | 48.014 ± 6.346 | | | |
| Job | Employed | 549 | 48.047±6.454 | -0.702 | 0.299 | 0.019 |
| | Housewife | 1968 | 49.200±6.018 | | | |
| Mother's ethnicity | Fars | 1712 | 49.388±5.982 | 0.975 | 0.279 | <0.001 |
| | Other ethnicities | 805 | 48.014±6.346 | | | |
| **Quantitative variables** | **Mean ± SD** | | **Median (1st Qu, 3rd Qu)** | **B** | **S.E** | **P value** |
| Breastfeeding duration | 10.109±4.559 | | 10(7,13.33) | 0.082 | 0.031 | 0.009 |
| Menstruation start age | 13.569±2.020 | | 14(12,15) | 0.214 | 0.060 | <0.001 |
| Education years | 1.625±2.745 | | 0(0,3) | -0.158 | 0.049 | 0.001 |

women with other ethnicities (p-value<0.001). In this population, most of the women were illiterate or poorly educated. Regression also confirmed that education years is adversely related to ANM (p-value = 0.001). We found the mean±SD of the first menstruation 13.57 ±2.02 years. With each year increase in the age of first menstruation, the ANM increases by 0.21 year (p-value<0.001). The mean±SD duration of breastfeeding was 10.11±4.56 years, which had a positive association with ANM (p-value = 0.009). Although breastfeeding has a significant relationship with ANM, every year increase in breastfeeding, ANM would increase only by 0.082 years or almost one month. We split the number of pregnancies based on the median (7 pregnancies), which is one of the most common methods for splitting [31]. Women who had more than 7 pregnancies had 1.065 years later menopause (p-value<0.001). Further analyses confirm the results of multiple regression; we found a correlation between the ANM and the number of pregnancies (r = 0.132, p<0.001), breastfeeding duration (r = 0.137, p<0.001), and menstruation start age (r = 0.093, p<0.001). Given the results of multiple linear regression, women with the experience of stillbirth had earlier menopause by 0.88 year (p-value = 0.005). Only 1.87% of women had used hormonal replacement drugs; they had 1.94 years earlier menopause. Actually, women who experienced earlier menopause were treated with hormone replacement drugs (p-value = 0.03).

## Conditional tree

In our effort to predict the determinants of menopause age according to fertility factors, a conditional inference tree was used (Fig 1). The tree found the number of pregnancies as the most important factor related to ANM. Other associated factors in order of their importance are as follows: breastfeeding duration, job, residence, use of hormone replacement drugs, and mother's ethnicity; so that Fars mother's ethnicity increases the ANM by almost two years. Among women with a maximum of one pregnancy, the duration of breastfeeding is the second most important factor. As can be seen in Fig 1, women with the highest (50.70) and the lowest ANM (41.22) are placed in nodes 7 and 10, respectively.

## Conditional forest

For a better understanding of predictors of menopause timing, we used Cforest to rank the variables according to their importance. As shown in Fig 2, the number of pregnancies,

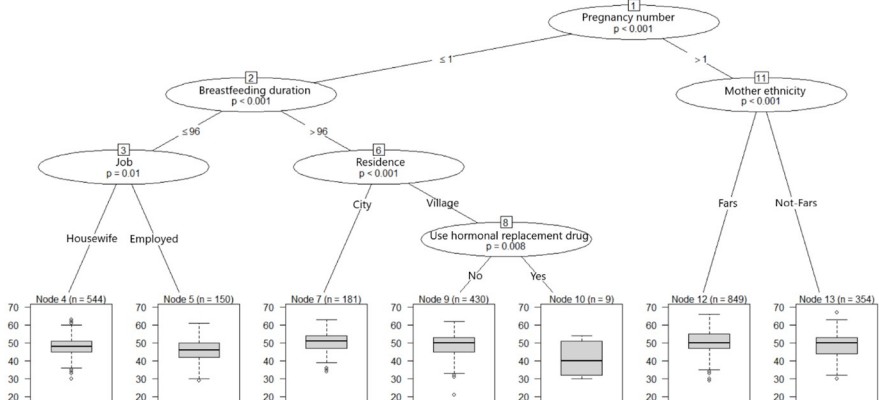

**Fig 1. Conditional inference tree grown on predictors of menopause age adapted from postmenopausal women of Kharameh.** At each terminal node, the box plot shows the distribution of menopause age.

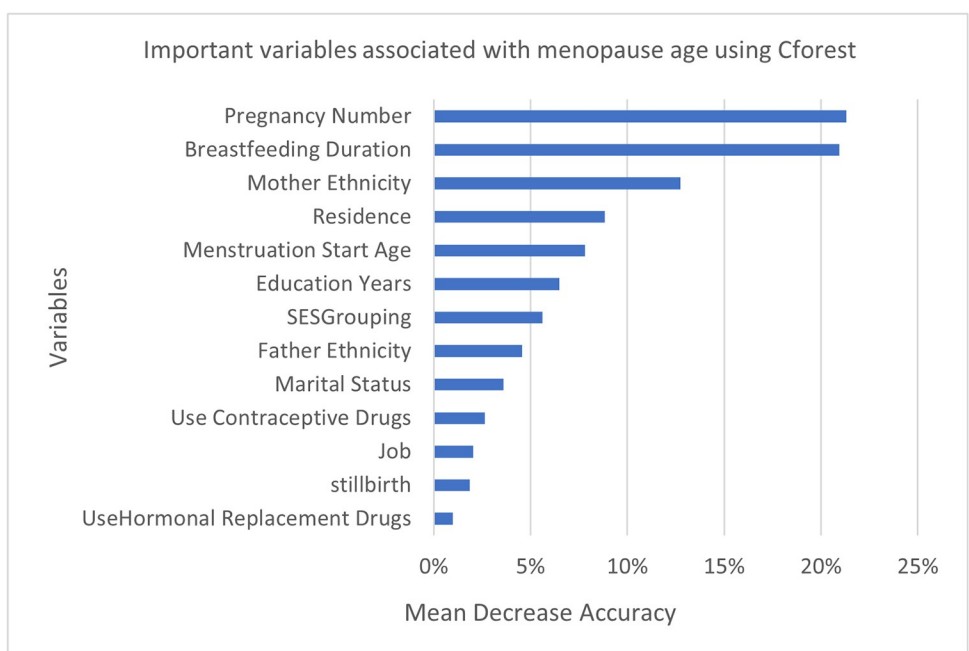

**Fig 2. Variable IMPortance associated with menopause age using Cforest among postmenopausal women of Kharameh.**

duration of breastfeeding, mother's ethnicity, and residence are the most important variables. In fact, 64% of correct predictions were due to these variables. Other predictors in order of their importance are as follows: menstruation starts age, education years, SES, father's ethnicity, marital status, use of contraceptive drugs, job, stillbirth, and use of hormonal replacement drugs.

## Random forest

Similar to Cforest, Rforest recognized the mother's ethnicity, number of pregnancies, breastfeeding duration, and residence as the most important predictors. However, in Rforest, 58% of correct predictions were due to these variables. According to Fig 3, other predictors in order of their importance are as follows: socioeconomic status, job, menstruation start age, marital status, stillbirth, education years, use of contraceptive drugs, use of hormonal replacement drugs, and father's ethnicity. It is worth mentioning that the father's ethnicity, marital status, and use of contraceptive drugs were simultaneously identified by Cforest and Rforest. Actually, Fars women, married ones, and contraceptive users had a later ANM.

## Comparison of methods

As we mentioned, we used two indices for comparing the goodness of fit of the applied methods regarding the ANM prediction and its related factors.

As shown in Table 2, MSE in the train set of Rforest is the lowest while in the test set, Cforest has the lowest amount of MSE. For comparing the methods according to the Pearson correlation coefficient, it was indicated that all the correlations between observed and predicted values were significant, by all methods and within both train and test sets. Based on this criterion, Rforest in the train set has the best performance while the performance of this method in the test set is similar to the Cforest. Therefore, in conclusion, according to the test set and by

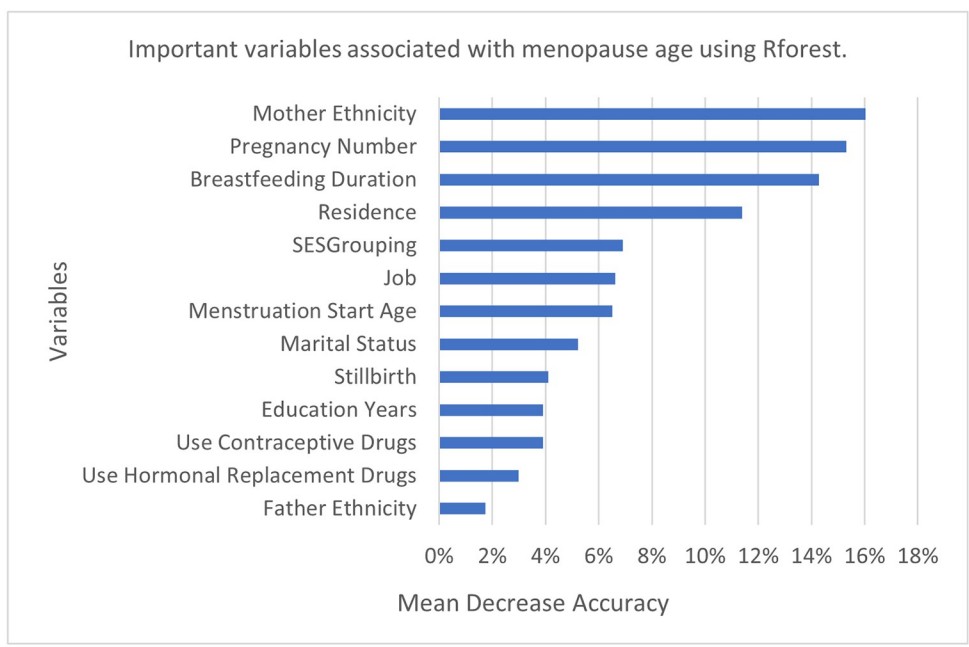

**Fig 3. Variable IMPortance associated with menopause age using Rforest among postmenopausal women of kharameh.**

considering both MSE and correlation coefficient, it can be claimed that Cforest's performance is the most accurate one. It is also valuable to note that there are many differences in Rforest's performance in two the train and test sets; In the way that mentioned indices indicated a much better fit in the train set in comparison to the test set. Therefore, it seems that the Rforest algorithm is overfitted in the train set.

## Discussion

In this population-based study on the evaluation of age at natural menopause and its determinants among post-menopausal women who participated in phase I Persian cohort Kharameh, we found the following: firstly, the mean±SD (95%CI) and median (1st Qu, 3rd Qu) of ANM were 48.95±6.13 (48.71, 49.19) and 50.00 (45.00, 53.00) years, respectively. Secondly, associated factors with ANM were: the number of pregnancies, duration of breastfeeding, mother's ethnicity, residence, menstruation start age, education years, SES, father's ethnicity, marital status, use of contraceptive drugs, occupation, stillbirth, and use of hormonal replacement drugs. Lastly, Cforest showed the best performance among other applied methods.

**Table 2. Comparison of 4 methods of regression, Ctree, Cforest, and Rforest applied for identifying the determinants of menopause age.**

| Method | MSE | | Pearson Correlation | | | |
|---|---|---|---|---|---|---|
| | Train | Test | Train | | Test | |
| | | | Coefficient | P value | Coefficient | P value |
| Regression | 35.705 | 35.090 | 0.257 | <0.001 | 0.188 | <0.001 |
| Ctree | 35.822 | 35.618 | 0.251 | <0.001 | 0.156 | <0.001 |
| Cforest | 29.776 | 34.785 | 0.576 | <0.001 | 0.196 | <0.001 |
| Rforest | 14.558 | 35.123 | 0.910 | <0.001 | 0.197 | <0.001 |

Compared to the other cities of Iran, the menopause age in this study is higher than Isfahan 48.66 [20], Shiraz 48.3 [32], Kerman 48.56 [33], and Gorgan 47.6 [34] and lower than Sari and Mazandaran 49.2 [18] and Hamadan 49.6 [35]. With a Global vision, it's higher than Turkey 47 [36], India 46.6 [37], Brazil 47.5 [38], China 48.8 [39] but still lower than developed countries like Taiwan 50.2 [40], Japan 50.4 [41], USA 52.6 [42], Spain 51.7 [42], France 52 [43] and Italy 51.2 [44].

The differences between our findings and previous results may be explained by sample size variations and differences in genetic background, lifestyle, and socioeconomic status in studied women.

As mentioned, we used four different methods for a better understanding of the determinants of menopause age. Associated Factors recognized by Ctree, were identified by other methods too; including number of pregnancies, breastfeeding duration, job, residence, mother's ethnicity, and use of hormone replacement drugs. Menstruation start age, socioeconomic status, education years, and stillbirth were detected in regression and both forests. Marital status, use of contraceptive drugs, and father's ethnicity were simultaneously identified by both forests.

Several studies have found a significant association between lower SES and lower education with earlier menopause [13, 20, 23, 45, 46]. In the current study, women with a better SES had later menopause, since it ranges from 48.36 years in a low socioeconomic background to 49.52 years in a very high socioeconomic background (Table 1). However, we found an inverse relationship between education years and ANM. Actually, there is an inverse correlation between the number of pregnancies and education years; in the way that people with more pregnancies are less educated. Due to the existence of education years and the number of pregnancies in a single regression model, it seems that the relationship between education years and ANM is confounded. Anyway, our results are in line with the findings of Sinha et al. [47] and Ismaeeli et al. [48]; that women with higher education and employment have an earlier menopause [47]. Actually, in our study, earlier menopause among employees may be because most of them (87.61%) are rural dwellers. As mentioned, rural women have an earlier menopause. However, these variables are not associated with ANM in some investigations [18, 21]. Another reason for the lower ANM in employed women may be due to their exposure to workplace stressors. Cassou et al. revealed that some job stressors might be associated with an increased risk of earlier age at menopause [43].

Previous investigations including a study conducted in Iran in 2013 found that menopause occurs earlier in rural areas than in urban areas [49–52]. Similarly, in this study, ANM in urban and rural areas was 49.39 and 48.01 years, respectively. A reason for the higher ANM in urban women can be their better socioeconomic status. So, socioeconomic status seems to be a confounder.

The effect of ethnicity/race has been investigated in several studies [53, 54]. For instance; Gold. E.B. et al. demonstrated that Japanese women have a later natural menopause than Caucasian, African-American, Hispanic, or Chinese women do [45]. According to another study, Japanese-American race/ethnicity was associated with late natural menopause and Latinas experienced early natural menopause, with those born in the United States experiencing menopause later than those born outside the United States [55]. The majority of the population of Kharamah are Fars and we observed that Fars ethnicity is a notable factor for later menopause. To the best of our knowledge, there is still no study that compares Fars ethnicity with other ethnicities. In addition, we considered both the father's and mother's ethnicity in our analysis. But against regression and Ctree which only identified the mother's ethnicity as a significant factor, both Cforest and Rforest, introduced the father's ethnicity as a less important factor next to the mother's ethnicity. According to the mother's ethnicity, the Fars group had almost

1 year later menopause. The mean of ANM was 49.39 and 48.01 for the Fars group and other ethnic group, respectively. This ethnic difference supports the hypothesis that ANM is a heritable trait. Torgerson et al. showed that there is a relationship between ANM and maternal ANM [56]. According to twin and family studies, the heritability score of ANM is 44% to 65% [57, 58].

Moreover, our further analysis showed that there is a significant correlation between Residency and the mother and father's ethnicity (p-value<0.001). So that Fars ethnic groups are more urban dwellers than rural dwellers and as mentioned, urban women have an older age at menopause. So, this may be another explanation for later menopause among Fars women. In summary, we found the ethnicity of the mother as a great predictor for menopause age which has not been considered in previous studies; perhaps girls mostly inherit menopause traits from their mothers. Therefore, more investigations are needed for a better understanding of this question.

Another reason for earlier ANM among rural women, women of other ethnicities, and women with a lower SES could be that these people have lower health literacy and less access to health providers [59, 60].

According to the regression model, breastfeeding has little impact on ANM and this statistical significance may be due to the large sample size but both forests identified it as one of the most important variables. However, in agreement with other studies [18, 22, 61], we found that longer breastfeeding and more pregnancies cause later menopause. Also, we observed that women with higher age at first menstruation have higher ANM which is consistent with some other studies [61–64]. Nonetheless, some studies have not shown a relationship [18]. Our results may be explained by the fixed follicle pool and therefore possibly a fixed number of ovulatory cycles. This mechanism may also explain the relationship between later menopause and more pregnancies, longer breastfeeding, and the use of hormonal contraceptives which all disturb the ovulation cycle [61]. Indeed, the onset of menopause is theorized to be related to the rate of loss of oocytes and thus to the occurrence of ovulatory cycles [47]. Past studies suggest that the use of hormonal contraceptives can delay menopause [61, 65–69], which has been proved by both Cforest and Rforest; although some studies have found no association [18].

Previous studies have mostly concentrated on abortion than stillbirth but we considered both of them. Although we didn't find any association between having abortions and ANM, we observed that women with the experience of stillbirth had an earlier menopause by 0.88 year. However, Dorjgochoo et al. found no relationship [68].

Hormone Replacement drugs contain the hormones that a woman's body stops producing after menopause. Actually, it is used to treat menopausal symptoms and it's recommended for women who experience early menopause until the average age of menopause [70]. We also found that women with younger menopause age had a history of HRT use.

According to Kamyabi et al., single women reach menopause earlier due to the regular activity of ovaries and successive stimulation of follicles, under the influence of pituitary hormones [71]. Both of our applied forests revealed that married women have a higher ANM, which is also consistent with some studies [19, 72]. However, it's in contrast with some other investigations [20, 25].

In summary, although the age of menopause is affected by non-modifiable factors such as genetics and ethnicity, some factors related to people's lifestyles are modifiable and should be considered in women's health programs.

Based on our research, this is the first study in which machine learning algorithms including Ctree and forests are used in addition to linear regression to analyze factors related to ANM. It should be declared that all applied methods confirm each other due to the overlap in identifying the factors related to ANM. Past studies have indicated that predictions provided

by Rforest are more accurate than Cforest [27, 73], although in another study conditional forest outperformed the regularized random forest [74]. However, given the comparison of these 4 methods using MSE and correlation coefficient, unlike Rforest which was overfitted in the train set, the performance of Cforest was much better than other methods in the test set. Furthermore, we found the Ctree as the least accurate method. In line with our results, the inconsistency and higher variance of tree estimates compared to forests [75–77] as well as logistic regression have been noted in previous studies [73]. Although in linear regression MSE was almost lower than Rforest, the correlation coefficient denotes the better performance of Rforest. So, our findings confirm the study of Ryu et al. [78], in which Rforest was more accurate than linear regression. however, in some other studies linear regression outperformed Rforest [79].

The strength of this population-based study is the large population of participants who entered the study through the census. Furthermore, we used four different methods for better prediction of factors affecting menopause age while three of these models have not been applied in Iranian society so far. This study has some limitations; it is a cross-sectional design, and the obtained relationships are not necessarily causal. In Addition, the ANM and other factors were self-reported by women; therefore, it depends on the participants to recall those events. Since the study group is women 40–70 years old, we were not able to consider postmenopausal women under 40 and over 70 years. Our study aimed to identify reproductive factors affecting menopause age by controlling some demographic variables. So, factors like genetics, physical activity, and illnesses have not been considered. More extensive and multi-center studies that examine all aforementioned variables simultaneously are needed.

## Supporting information

**S1 Checklist. Human participants research checklist.**
(DOCX)

## Acknowledgments

This article was a part of Zahra Pasokh's M.Sc. thesis.

## Author Contributions

**Conceptualization:** Zahra Pasokh, Mozhgan Seif, Haleh Ghaem, Abbas Rezaianzadeh.

**Data curation:** Zahra Pasokh, Abbas Rezaianzadeh, Masoumeh Ghoddusi Johari.

**Formal analysis:** Zahra Pasokh.

**Investigation:** Zahra Pasokh, Abbas Rezaianzadeh.

**Methodology:** Zahra Pasokh, Mozhgan Seif.

**Project administration:** Zahra Pasokh, Mozhgan Seif, Abbas Rezaianzadeh.

**Software:** Zahra Pasokh.

**Supervision:** Mozhgan Seif, Haleh Ghaem.

**Validation:** Mozhgan Seif.

**Writing – original draft:** Zahra Pasokh.

**Writing – review & editing:** Mozhgan Seif, Haleh Ghaem, Abbas Rezaianzadeh.

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
