## [Decision Letter · Decision Letter 0]

14 Jan 2024

PONE-D-23-34559Age at Natural Menopause and Its Determinants in Female Population of Kharameh Cohort Study: Comparison of Regression, Conditional Tree and ForestsPLOS ONE

Dear Dr. Seif,

Thank you for submitting your manuscript to PLOS ONE. After careful consideration, we feel that it has merit but does not fully meet PLOS ONE’s publication criteria as it currently stands. Therefore, we invite you to submit a revised version of the manuscript that addresses the points raised during the review process.

**ACADEMIC EDITOR: **

We look forward to receiving your revised manuscript.

Kind regards,

Zahra Cheraghi, Ph.D

Academic Editor

PLOS ONE

“This article was a part of Zahra Pasokh’s M.Sc. thesis approved and financially supported by the Research Vice-chancellor of Shiraz University of Medical Sciences (grant No. 25007).”

“This article was a part of Zahra Pasokh’s M.Sc. thesis approved and financially supported by the Research Vice-chancellor of Shiraz University of Medical Sciences (grant No. 25007). This study was approved by the Ethics Committee of Shiraz University of Medical Sciences (IR.SUMS.SCHEANUT.REC.1400.104).”

“This article was a part of Zahra Pasokh’s M.Sc. thesis approved and financially supported by the Research Vice-chancellor of Shiraz University of Medical Sciences (grant No. 25007).”

5. In the online submission form you indicate that your data is not available for proprietary reasons and have provided a contact point for accessing this data. Please note that your current contact point is a co-author on this manuscript. According to our Data Policy, the contact point must not be an author on the manuscript and must be an institutional contact, ideally not an individual. Please revise your data statement to a non-author institutional point of contact, such as a data access or ethics committee, and send this to us via return email. Please also include contact information for the third party organization, and please include the full citation of where the data can be found.

Reviewers' comments:

Reviewer's Responses to Questions

**Comments to the Author**

1. Is the manuscript technically sound, and do the data support the conclusions?

Reviewer #1: Yes

Reviewer #2: Yes

2. Has the statistical analysis been performed appropriately and rigorously? 

Reviewer #1: Yes

Reviewer #2: No

3. Have the authors made all data underlying the findings in their manuscript fully available?

Reviewer #1: Yes

Reviewer #2: Yes

4. Is the manuscript presented in an intelligible fashion and written in standard English?

Reviewer #1: Yes

Reviewer #2: Yes

5. Review Comments to the Author

Reviewer #1: This is an interesting, well-written article with excellent methodology that examines factors affecting ANM using various machine learning methods. In my opinion, this article is suitable for publication in PLoS ONE, just correct the following points in the modified version.

1. What is meant by natural menopause? Please explain it in the abstract section.

2. The authors wrote in the introduction that ethnicity, region, socioeconomic position, environment, life style, occupation, education, residence, and marital status have an effect on menopause. Please specify which subgroups of these variables have a positive effect on premature menopause.

3. The phrase " As can be seen, breastfeeding has little impact on ANM and this statistical significance may be due to the large sample size." should be transferred to the discussion in the findings section.

Reviewer #2: This study investigated the age at Natural Menopause and Its Determinants in Female Population of Kharameh Cohort Study: Comparison of Regression, Conditional Tree and Forests. The topic is interesting. The topic is interesting. However, there are some concerns that need to be addressed.

Abstract

Results:

All methods simultaneously recognized the number of pregnancies, duration of breastfeeding, mother’s ethnicity, job, and residence as the most important predictors of ANM.

-The sentence is ambiguous and the direction of association is unclear. For example, it is not clear whether the association is positive or negative. For instance, it is unclear whether increasing the frequency of pregnancy lowers or increases the age of menopause. Similarly, an unclear and vague relationship has been reported for other variables.

Methods

-Is the age range of 40-70 years appropriate and based on a reliable reference? Typically, the onset of menopause occurs between the ages of 45 and 55. Could you please provide the reference used to select this age group?

-What was the response rate in this study?

Discussion

-In this study, the cross-sectional phase was used. Therefore, the relationships obtained are not necessarily causal. It is important to explain these relationships with caution and to include them in the limitations section.

6. PLOS authors have the option to publish the peer review history of their article (what does this mean?). If published, this will include your full peer review and any attached files.

Reviewer #1: **Yes: **Fatemeh Shahbazi

Reviewer #2: No

---

## [Author Response · Author response to Decision Letter 0]

21 Feb 2024

Dear Editor,

We would like to express our gratitude for the opportunity to submit a revised draft of the manuscript. We sincerely appreciate the time and effort you and the reviewers have dedicated to providing feedback on our work. After carefully reviewing the comments, we have made the necessary revisions to the manuscript, which are highlighted within the document. We have also addressed the reviewers' comments point by point below. With the implemented changes, we believe that the revised version is now suitable for publication. We eagerly await your response.

Thank you once again for your consideration.

Sincerely, Mozhgan Seif

Author's response: Done, we applied the requirements in the manuscript. We added Author contributions before the Abstract, used the level 1 heading for major sections, and level 2 heading for sub-sections of major sections, abbreviated the word “Figure” to “Fig” and placed figures after the paragraph in which they are first cited, and used bold type for figures titles. We placed tables after the paragraph in which they are first cited, and used bold type for table titles. We added page number and page line, cited references in brackets, and removed competing interests and data availability statements at the end of the manuscript.

Author's response: Dr. Abbas Rezaianzadeh is the principal responsible for the Kharameh cohort study. Although he is a co-author of this manuscript, he doesn’t allow the data to be publicly available. However, he provides the data to researchers under special circumstances by corresponding with him. (Contact via: rezaiana@gmail.com)

“This article was a part of Zahra Pasokh’s M.Sc. thesis approved and financially supported by the Research Vice-chancellor of Shiraz University of Medical Sciences (grant No. 25007).”

Author's response: the funders had no role in this research. We edited the funding statement and moved the funding statement to the cover letter.

“This article was a part of Zahra Pasokh’s M.Sc. thesis approved and financially supported by the Research Vice-chancellor of Shiraz University of Medical Sciences (grant No. 25007). This study was approved by the Ethics Committee of Shiraz University of Medical Sciences (IR.SUMS.SCHEANUT.REC.1400.104).”

“This article was a part of Zahra Pasokh’s M.Sc. thesis approved and financially supported by the Research Vice-chancellor of Shiraz University of Medical Sciences (grant No. 25007).”

Author's response: done, we omitted the funding information from the acknowledgments section. Actually, we removed the funding information from any part of the manuscript and referenced it in the cover letter.

5. In the online submission form you indicate that your data is not available for proprietary reasons and have provided a contact point for accessing this data. Please note that your current contact point is a co-author on this manuscript. According to our Data Policy, the contact point must not be an author on the manuscript and must be an institutional contact, ideally not an individual. Please revise your data statement to a non-author institutional point of contact, such as a data access or ethics committee, and send this to us via return email. Please also include contact information for the third party organization, and please include the full citation of where the data can be found.

Author's response: Dr. Abbas Rezaianzadeh is the principal responsible for the Kharameh cohort study. Although he is a co-author of this manuscript, he doesn’t allow the data to be publicly available. However, he provides the data to researchers under special circumstances by corresponding with him. (Contact via: rezaiana@gmail.com)

Author's response: We had mentioned the ethics statement in the acknowledgment section. According to your comment, we removed it from acknowledgment and mentioned it in the methods section (page 6 lines 121-124).

Reviewer #1:

This is an interesting, well-written article with excellent methodology that examines factors affecting ANM using various machine learning methods. In my opinion, this article is suitable for publication in PLoS ONE, just correct the following points in the modified version.

1. What is meant by natural menopause? Please explain it in the abstract section.

Author's response: done, thank you for your suggestion. Natural menopause is defined as the permanent cessation of menstruation that occurs after 12 consecutive months of amenorrhea without any obvious pathological or physiological cause. This type of menopause is defined in contrast to induced menopause. Induced menopause is the menopause that occurs after surgery to remove the ovaries, chemotherapy or radiation damage to the ovaries, or the use of other drugs to intentionally induce menopause as part of the treatment of certain diseases. We added definition of natural menopause in the abstract (page 1).

2. The authors wrote in the introduction that ethnicity, region, socioeconomic position, environment, life style, occupation, education, residence, and marital status have an effect on menopause. Please specify which subgroups of these variables have a positive effect on premature menopause.

Author's response: done (page 4 lines 79-87). Thank you for your suggestion.

3. The phrase " As can be seen, breastfeeding has little impact on ANM and this statistical significance may be due to the large sample size." should be transferred to the discussion in the findings section.

Author's response: corrected (page 18, lines 357-358), thank you.

Reviewer #2:

 This study investigated the age at Natural Menopause and Its Determinants in Female Population of Kharameh Cohort Study: Comparison of Regression, Conditional Tree and Forests. The topic is interesting. The topic is interesting. However, there are some concerns that need to be addressed.

Abstract

Results:

All methods simultaneously recognized the number of pregnancies, duration of breastfeeding, mother’s ethnicity, job, and residence as the most important predictors of ANM.

-The sentence is ambiguous and the direction of association is unclear. For example, it is not clear whether the association is positive or negative. For instance, it is unclear whether increasing the frequency of pregnancy lowers or increases the age of menopause. Similarly, an unclear and vague relationship has been reported for other variables.

Author's response: corrected (page 2), thank you.

Methods

-Is the age range of 40-70 years appropriate and based on a reliable reference? Typically, the onset of menopause occurs between the ages of 45 and 55. Could you please provide the reference used to select this age group?

Author's response: thanks for our question. As you mentioned, natural menopause normally occurs between the ages of 45 and 55 but some people may experience natural menopause earlier or later of this age group. We have used the data collected in the Kharameh cohort study. The Kharameh cohort study is a comprehensive study conducted on people aged 40-70 years old. This study aimed to determine the factors which may affect age at natural menopause. Therefore, considering the availability of data, the age range considered in this study is 40 to 70 years old.

-What was the response rate in this study?

Author's response: as we have mentioned in page 5, out of 10,836 people, 10,663 people (5,944 women and 4,719 men), which is equivalent to 98.4% of the population of Kharamah, participated in this census and answered questioners. Out of 5944 women, 2517 people were reported to be naturally menopause. So, we used these women’s information for our study which all were completed the questioners. (page 5 lines 113-115, page 6 line 119-120)

Discussion

-In this study, the cross-sectional phase was used. Therefore, the relationships obtained are not necessarily causal. It is important to explain these relationships with caution and to include them in the limitations section.

Author's response: thank you, we applied your suggestion in limitations (page 20-21 lines 403-404).

---

## [Editor Report · Decision Letter 1]

28 Feb 2024

Age at Natural Menopause and Its Determinants in Female Population of Kharameh Cohort Study: Comparison of Regression, Conditional Tree and Forests

PONE-D-23-34559R1

Dear Dr. Seif,

We’re pleased to inform you that your manuscript has been judged scientifically suitable for publication and will be formally accepted for publication once it meets all outstanding technical requirements.

Kind regards,

Zahra Cheraghi, Ph.D

Academic Editor

PLOS ONE
---

## [Editor Report · Acceptance letter]

4 Apr 2024

PONE-D-23-34559R1 

PLOS ONE

Dear Dr. Seif, 

I'm pleased to inform you that your manuscript has been deemed suitable for publication in PLOS ONE. Congratulations! Your manuscript is now being handed over to our production team.

Kind regards, 

on behalf of

Dr. Zahra Cheraghi 

Academic Editor

PLOS ONE